# O-GlcNAc glycosylation orchestrates fate decision and niche function of bone marrow stromal progenitors

**Zengdi Zhang**[1†], **Zan Huang**[1,2,3†], **Mohamed Awad**[4], **Mohammed Elsalanty**[4], **James Cray**[5], **Lauren E Ball**[6], **Jason C Maynard**[7], **Alma L Burlingame**[7], **Hu Zeng**[8,9], **Kim C Mansky**[10], **Hai-Bin Ruan**[1,11]*

[1]Department of Integrative Biology and Physiology, University of Minnesota Medical School, Minneapolis, United States; [2]Laboratory of Gastrointestinal Microbiology, Jiangsu Key Laboratory of Gastrointestinal Nutrition and Animal Health, College of Animal Science and Technology, Nanjing Agricultural University, Nanjing, China; [3]National Center for International Research on Animal Gut Nutrition, Nanjing Agricultural University, Nanjing, China; [4]Department of Medical Anatomical Sciences, College of Osteopathic Medicine of the Pacific, Western University of Health Sciences, Pomona, United States; [5]Department of Biomedical Education and Anatomy, The Ohio State University College of Medicine, and Division of Biosciences, The Ohio State University College of Dentistry, Columbus, United States; [6]Department of Cell and Molecular Pharmacology & Experimental Therapeutics, Medical University of South Carolina, Charleston, United States; [7]Department of Pharmaceutical Chemistry, University of California, San Francisco, San Francisco, United States; [8]Division of Rheumatology, Department of Internal Medicine, Mayo Clinic, Rochester, United States; [9]Department of Immunology, Mayo Clinic, Rochester, United States; [10]Department of Developmental and Surgical Sciences, School of Dentistry, University of Minnesota, Minneapolis, United States; [11]Center for Immunology, University of Minnesota Medical School, Minneapolis, United States

**\*For correspondence:**
hruan@umn.edu

†These authors contributed equally to this work

**Competing interest:** The authors declare that no competing interests exist.

**Summary** In mammals, interactions between the bone marrow (BM) stroma and hematopoietic progenitors contribute to bone-BM homeostasis. Perinatal bone growth and ossification provide a microenvironment for the transition to definitive hematopoiesis; however, mechanisms and interactions orchestrating the development of skeletal and hematopoietic systems remain largely unknown. Here, we establish intracellular O-linked β-N-acetylglucosamine (O-GlcNAc) modification as a post-translational switch that dictates the differentiation fate and niche function of early BM stromal cells (BMSCs). By modifying and activating RUNX2, O-GlcNAcylation promotes osteogenic differentiation of BMSCs and stromal IL-7 expression to support lymphopoiesis. In contrast, C/EBPβ-dependent marrow adipogenesis and expression of myelopoietic stem cell factor (SCF) is inhibited by O-Glc-NAcylation. Ablating O-GlcNAc transferase (OGT) in BMSCs leads to impaired bone formation, increased marrow adiposity, as well as defective B-cell lymphopoiesis and myeloid overproduction in mice. Thus, the balance of osteogenic and adipogenic differentiation of BMSCs is determined by reciprocal O-GlcNAc regulation of transcription factors, which simultaneously shapes the hematopoietic niche.

## Editor's evaluation

Bone marrow stromal cells (BMSCs) can differentiate into a variety of cell types such as osteoblasts, chondrocytes, and adipocytes. The authors of this important study provide compelling and strong evidence that ablating O-GlcNAc transferase (OGT) in BMSCs impairs bone formation but promotes marrow adiposity. The results show that the balance of osteogenic and adipogenic differentiation of BMSCs is controlled by reciprocal O-GlcNAc regulation of lineage-specifying transcription factors, and highlights the importance of an intracellular glycosylation process of specific proteins in establishing the BM niche for hematopoiesis.

## Introduction

Mammalian bones support body structure, protect vital organs, and allow body movement. In addition, they provide an environment for hematopoiesis in the bone marrow (BM). Most bones in mammals are formed through endochondral ossification, which is initiated by mesenchymal condensation, followed by the differentiation of chondrocytes and perichondrial progenitors (*Kobayashi and Kronenberg, 2021*). Perichondrial progenitors expressing Osterix (encoded by the *Sp7* gene) co-migrate with blood vessels into the primary ossification center, giving rise to osteoblasts and transient stromal cells in the nascent BM cavity (*Chen et al., 2014*; *Liu et al., 2013*; *Maes et al., 2010*; *Mizoguchi et al., 2014*). At the perinatal stage, Osterix[+] progenitors contribute to osteo-lineages and long-lived BM stromal cells (BMSCs) that exhibit trilineage differentiation potential into osteocytes, chondrocytes, and adipocytes.

The decision of BMSC fate is controlled by a transcriptional network of pro-osteogenic and anti-adipogenic transcription factors that pre-establishes osteogenic enhancers in BMSCs for rapid bone formation (*Rauch et al., 2019*). RUNX family transcription factor 2 (RUNX2), by regulating osteogenic genes including *Sp7*, determines the osteoblast lineage from the multipotent BMSCs. Mice with *Runx2* mutations completely lack skeletal ossification and die of respiratory failure (*Komori et al., 1997*). *Runx2*-haploinsufficient mice show specific skeletal abnormalities characteristic of human cleidocranial dysplasia (CCD), including persistent fontanels, delayed closure of cranial sutures, rudimentary clavicles, and dental abnormalities (*Otto et al., 1997*; *Takarada et al., 2016*). On the other hand, adipogenesis is driven by downregulation of pro-osteogenic factors, remodeling of the chromatin, and activation of adipogenic transcription factors, such as C/EBPs and PPARγ (*Aaron et al., 2021*; *Rauch et al., 2019*). BM adiposity is associated with bone loss in osteoporosis caused by aging, menopause, and anorexia nervosa (*Bethel et al., 2013*; *Fazeli et al., 2013*; *Liu et al., 2015*; *Scheller et al., 2016*). However, it is incompletely understood how these distinct types of transcription factors act cooperatively to determine lineage differentiation during neonatal skeletal development.

BMSCs and their lineage-differentiated progeny (e.g. osteoblasts and adipocytes) provide a niche microenvironment for hematopoiesis (*Bianco and Robey, 2015*; *Calvi and Link, 2015*; *Morrison and Scadden, 2014*; *Wei and Frenette, 2018*). Recent studies using single-cell technologies and lineage tracing experiments have started to unveil the complexity and heterogeneity of niche cell types, niche factors, and their actions. For example, BMSC-derived stem cell factor (SCF, encoded by the *Kitl* gene) and CXC chemokine ligand 12 (CXCL12) are required for the maintenance and differentiation of hematopoietic stem/progenitor cells (HSPCs) (*Asada et al., 2017*; *Ding et al., 2012*). A prominent subpopulation of perivascular BMSCs express adipocyte markers (*Dolgalev and Tikhonova, 2021*; *Zhong et al., 2020*; *Zhou et al., 2017*), and support steady-state and metabolic-stressed myelopoiesis by secreting SCF (*Zhang et al., 2019*). Meanwhile, osteolineage cells are crucial for lymphopoiesis (*Wei and Frenette, 2018*). Depleting Osterix[+] cells halts B cell maturation and causes immune failure (*Yu et al., 2016*). IL-7, the most crucial factor for lymphoid progenitors, is expressed by a subset of BMSCs (*Fistonich et al., 2018*). While it is well accepted that myeloid and lymphoid progenitors may reside in distinct BM niches, it is unclear how BMSC heterogeneity is established during early development and whether cytokine expression is coordinated and controlled by the fate-defining transcriptional network in BMSCs.

Post-translational modifications (PTMs), including phosphorylation, acetylation, and ubiquitination, allow the precise regulation of stability, localization, and activity of BM transcriptional factors, such as RUNX2 (*Chen et al., 2021*; *Kim et al., 2020*), C/EBPs (*Wang et al., 2022*), and PPARγ (*Brunmeir and Xu, 2018*). It remains poorly defined how these modifications are coordinated in a spatio-temporal

manner to calibrate skeletal development. Thousands of intracellular proteins are dynamically modified by a single O-linked N-Acetylglucosamine (O-GlcNAc) moiety at serine or threonine residues, termed O-GlcNAcylation (*Hart et al., 2007*; *Ruan et al., 2013b*; *Yang and Qian, 2017*). O-GlcNAc transferase (OGT), using UDP-GlcNAc derived from the hexosamine biosynthetic pathway as the substrate, controls diverse biological processes such as gene transcription, protein stability, and cell signaling (*Hanover et al., 2012*; *Ruan et al., 2014*; *Ruan et al., 2012*; *Ruan et al., 2013a*). In cell culture, O-GlcNAcylation promotes osteogenesis (*Kim et al., 2007*; *Nagel and Ball, 2014*) and suppresses adipogenesis (*Ji et al., 2012*). However, the physiological relevance of O-GlcNAcylation in skeletal development and remodeling has not been established. Here, we studied OGT in balancing osteogenic versus adipogenic programs and in controlling niche function of BMSC in mice. The multifaceted role of protein O-GlcNAcylation is achieved through reciprocal regulation of pro-osteogenic, pro-lymphopoietic RUNX2 and pro-adipogenic, pro-myelopoietic C/EBPβ.

## Results

### Loss of OGT in perinatal BMSCs leads to bone loss

To determine the in vivo role of protein O-GlcNAcylation in bone development, we deleted the X Chromosome-located *Ogt* gene using the *Sp7*$^{GFP:Cre}$ mice (*Figure 1A*). Floxed *Ogt*$^{fl/fl}$ were bred with *Sp7*$^{GFP:Cre}$ to generate *Ogt* conditional knockout (cKO) mice. Compared to *Sp7*$^{GFP:Cre}$ littermate controls, newborn *Ogt* cKO mice showed no obvious change in long bone formation (*Figure 1B*) but had a profound defect in the mineralization of flat bones of the calvaria (*Figure 1C*), suggesting impaired intramembranous ossification during the prenatal stage.

At 4–6 weeks of age, *Ogt* cKO mice were modestly shorter than the controls (*Figure 1D and E*). Histological analyses showed decreased bone volume and osteoblast number (*Figure 1F*) and shortened growth plate (*Figure 1G*) in *Ogt* cKO mice. Micro-CT scanning further showed that *Ogt* cKO mice had reduced trabecular bone volume, bone volume to tissue volume ratio, trabecular thickness, and trabecular numbers in the distal femur (*Figure 1H–L*). *Ogt* cKO mice represent typical bone and dental defects (*Figure 1—figure supplement 1*) as observed in *Runx2*-haploinsufficient mice (*Otto et al., 1997*; *Takarada et al., 2016*), suggesting that O-GlcNAcylation might control RUNX2 function. Moist food was provided to these animals after weaning to prevent malnutrition.

### RUNX2 O-GlcNAcylation promotes osteogenesis

To investigate how OGT controls osteogenic differentiation of BMSCs, we first isolated primary BMSCs from control and *Ogt* cKO mice and induced them into osteoblast cells. Alkaline phosphatase staining revealed a reduction in mineralization of *Ogt* cKO BMSCs (*Figure 2A*). Similarly, treating mesenchymal C3H10T1/2 cells with an OGT inhibitor, OSMI-1, reduced mineralization (*Figure 2B*) and ablated calcium deposition (*Figure 2C*) after osteogenic differentiation. Parathyroid hormone (PTH) is a bone anabolic agent that requires RUNX2-dependent signaling (*Krishnan et al., 2003*). We found that PTH treatment of C3H10T1/2 cells increased global protein O-GlcNAcylation (*Figure 2D*). The ability of PTH to activate osteogenesis is completely abolished when OGT was inhibited by OSMI-1 (*Figure 2E*). Pharmacological activation of O-GlcNAcylation enhances RUNX2 activity and promotes osteogenic differentiation (*Kim et al., 2007*; *Nagel and Ball, 2014*). We mutated three known O-GlcNAc sites on RUNX2, Ser 32 and Ser 33 in the N-terminal transactivation domain and Ser 371 in the proline/serine/threonine-rich domain (*Figure 2F*), to alanine (3A), and found that mutant RUNX2 possessed less O-GlcNAcylation (*Figure 2G*). O-GlcNAcase (OGA) inhibition by Thiamet-G (TMG) increased O-GlcNAcylation of wildtype (WT) RUNX2, but to a much less extent in the 3A mutant (*Figure 2G*). OGT inhibition by OSMI-1 or O-GlcNAc mutation both impaired the transcriptional activity of RUNX2 on a luciferase reporter (*Figure 2H*). OSMI-1 could still suppress of luciferase activity of RUNX2-3A (*Figure 2H*), suggesting additional, unidentified O-GlcNAc sites (*Figure 2G*), which requires future investigation. Nevertheless, when overexpressed in C3H10T1/2 cells, RUNX2-3A substantially lost the ability to induce osteogenic differentiation (*Figure 2I*) or RUNX2-target gene expression (*Figure 2J*). These data demonstrate that O-GlcNAcylation is essential for RUNX2 activity and osteogenesis.

In adult mice, *Sp7* expression is restricted to osteoblast precursors. We treated *Ogt* cKO mice from pregnancy with doxycycline (Dox) and withdrew Dox at 10 weeks of age to induce Cre expression and OGT depletion only during adulthood (*Figure 3A*). Micro-CT showed that *Ogt* cKO mice had reduced

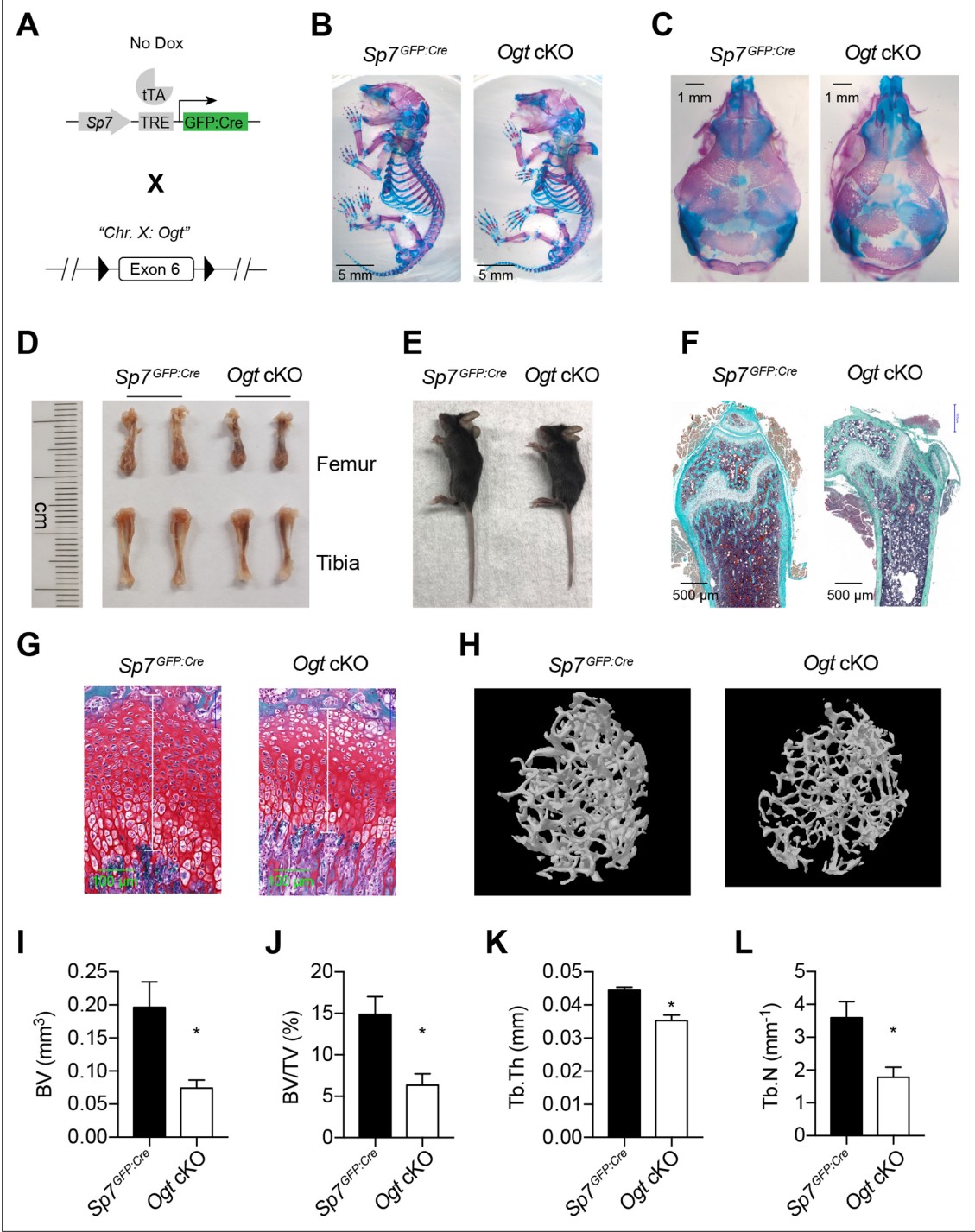

**Figure 1.** Impaired osteogenesis in *Ogt* cKO mice. (**A**) Mating strategy to generate *Ogt* cKO mice. Note that the *Ogt* gene is located on Chr. X, thus males are hemizygous *Ogt*<sup>fl/Y</sup>. (**B, C**) Whole mount Alizarin red and Alcian blue staining of newborn mice. (**D, E**) Long bone length (**D**) and gross morphology of 4–6 weeks old mice. (**F, G**) Goldner's trichrome (**F**) and Safranin O (**G**) staining of femurs from 4-week-old mice. (**H–L**) Micro-CT of 6-week-old mice (**H**, n=3–4). Bone volume (BV, **I**), BV/tissue volume ratio (BV/TV, **J**), trabecular thickness (Tb.Th, **K**), and trabecular number (Tb.N, **L**) were calculated. Data are presented as mean ± SEM. *, p<0.05 by unpaired student's *t*-test.

The online version of this article includes the following figure supplement(s) for figure 1:

**Figure supplement 1.** Dental defects of *Ogt* cKO mice.

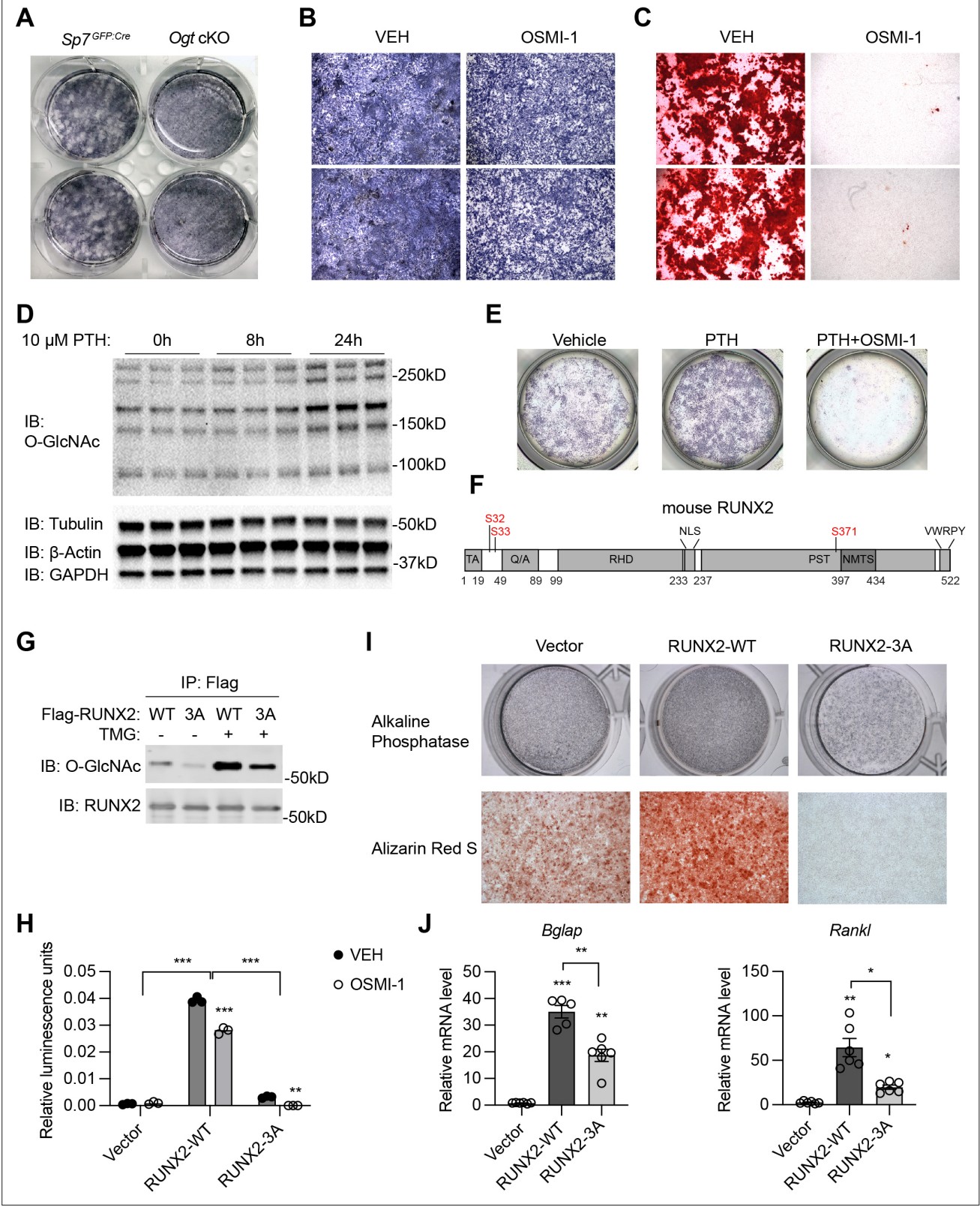

**Figure 2.** RUNX2 O-GlcNAcylation is required for osteogenesis. (**A**) Alkaline phosphatase (ALP) staining of control and *Ogt* cKO BMSCs differentiated to the osteogenic lineage. (**B, C**) Primary BMSCs, in the presence or absence of the OGT inhibitor OSMI-1, were induced for osteogenesis and stained for ALP (**B**) and Alizarin Red S (**C**). (**D**) Primary BMSCs were treated with PTH for the indicated time and subjected to Western blotting of total protein O-GlcNAcylation. (**E**) BMSCs were treated with PTH alone or together with OSMI-1, osteogenic differentiated, and stained for ALP (n=3). (**G**) Flag-tagged

*Figure 2 continued on next page*

*Figure 2 continued*

wildtype (WT) and O-GlcNAc mutant (3A) RUNX2 plasmids were overexpressed in HEK293 cells, and their O-GlcNAcylation was determined by Flag immunoprecipitation followed with O-GlcNAc western blot. (**H**) 6xOSE-luciferase activity in COS-7 cells transfected with WT or 3A-mutant RUNX2, in the presence or absence of the OGT inhibitor, OSMI-1. (**I, J**) C3H10T1/2 cells with lentiviral overexpression of RUNX2 were osteogenically differentiated and stained with ALP or Alizarin Red S (**I**). Expression of *Bglap* and *Rankl* was determined by RT-qPCR (**J**). Data are presented as mean ± SEM. *, p<0.05; **, p<0.01; and ***, p<0.001 by two-way ANOVA (**H**) or one-way ANOVA (**J**). Representative images from at least three biological replicates were shown in A, B, C, E, and I.

The online version of this article includes the following source data for figure 2:

**Source data 1.** Raw uncropped images for panel D.

**Source data 2.** Raw uncropped images for panel G.

bone volume, trabecular thickness, and bone mineral density (*Figure 3B–E*). Together, these results support the functional indispensability of OGT in the committed osteolineage for adult trabecular bone remodeling.

## C/EBPβ O-GlcNAcylation inhibits the adipogenic specification of BMSCs

The osteogenic and adipogenic differentiation of BMSCs is generally considered mutually exclusive (*Ambrosi et al., 2017*). Concomitant with bone loss, we observed a massive accumulation of adipocytes in the bone marrow of *Ogt* cKO mice, shown by hematoxylin & eosin staining (*Figure 4A*) and immune-staining of the lipid droplet protein – perilipin (*Figure 4B*). *Pdgfrα* and *Vcam1* (encoding CD106) have been recently identified as surface markers of adipogenic lineage cells in the BM that also express the *Lepr* and *Adipoq* genes (*Figure 4—figure supplement 1*; *Baryawno et al., 2019*; *Zhong et al., 2020*). Flow cytometric analysis of BMSCs revealed that *Ogt* cKO mice possessed more PDGFRα⁺VCAM1⁺ adipogenic progenitors than littermate controls (*Figure 4C*). To directly test if OGT deficiency biases BMSC differentiation toward the adipogenic lineage, we first induced the adipogenic differentiation of primary BMSCs and found increased lipid deposition in *Ogt* cKO mice (*Figure 4D*). Even under an osteogenic induction condition, adipo-lineage markers such as *Adipoq* and *Vcam1* were significantly upregulated by OGT deficiency (*Figure 4E and F*). Furthermore, treating C3H10T1/2 mesenchymal cells with an OGA inhibitor TMG to increase protein O-GlcNAcylation, was able to substantially reduce perilipin protein expression (*Figure 4G*) and *Pparg* and *Adipoq* gene expression (*Figure 4H, I*). These data indicate that OGT inhibits the adipogenic program of BMSCs.

We went on to determine the O-GlcNAc targets of OGT in suppressing adipogenesis. As an osteogenic regulator, RUNX2 also reciprocally suppresses the adipogenic program (*Ahrends et al., 2014*). However, such suppression was not dependent on O-GlcNAcylation, because O-GlcNAc-deficient RUNX2 displayed similar efficiency as the wildtype protein to reduce lipid deposition and perilipin expression in differentiated C3H10T1/2 cells (*Figure 4—figure supplement 2*). It is possible that O-GlcNAc on RUNX2 selectively facilitates the recruitment of transcriptional co-activators for

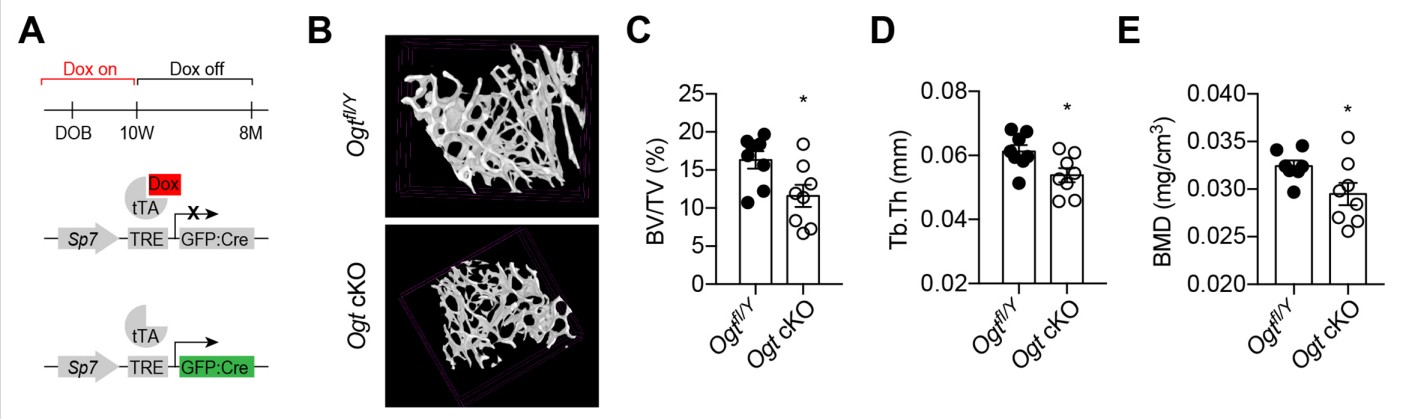

**Figure 3.** Adult-onset deletion of OGT impairs trabecular bone formation. (**A**) Dox treatment timeline in *Ogt* cKO to achieve osteoblast-specific deletion of OGT. (**B–D**) Micro-CT (**B**) showing reduced bone volume/tissue volume (**C**), trabecular thickness (**D**), and trabecular bone mineral density (**E**). Data are presented as mean ± SEM.*, p<0.05 by unpaired student's t-test.

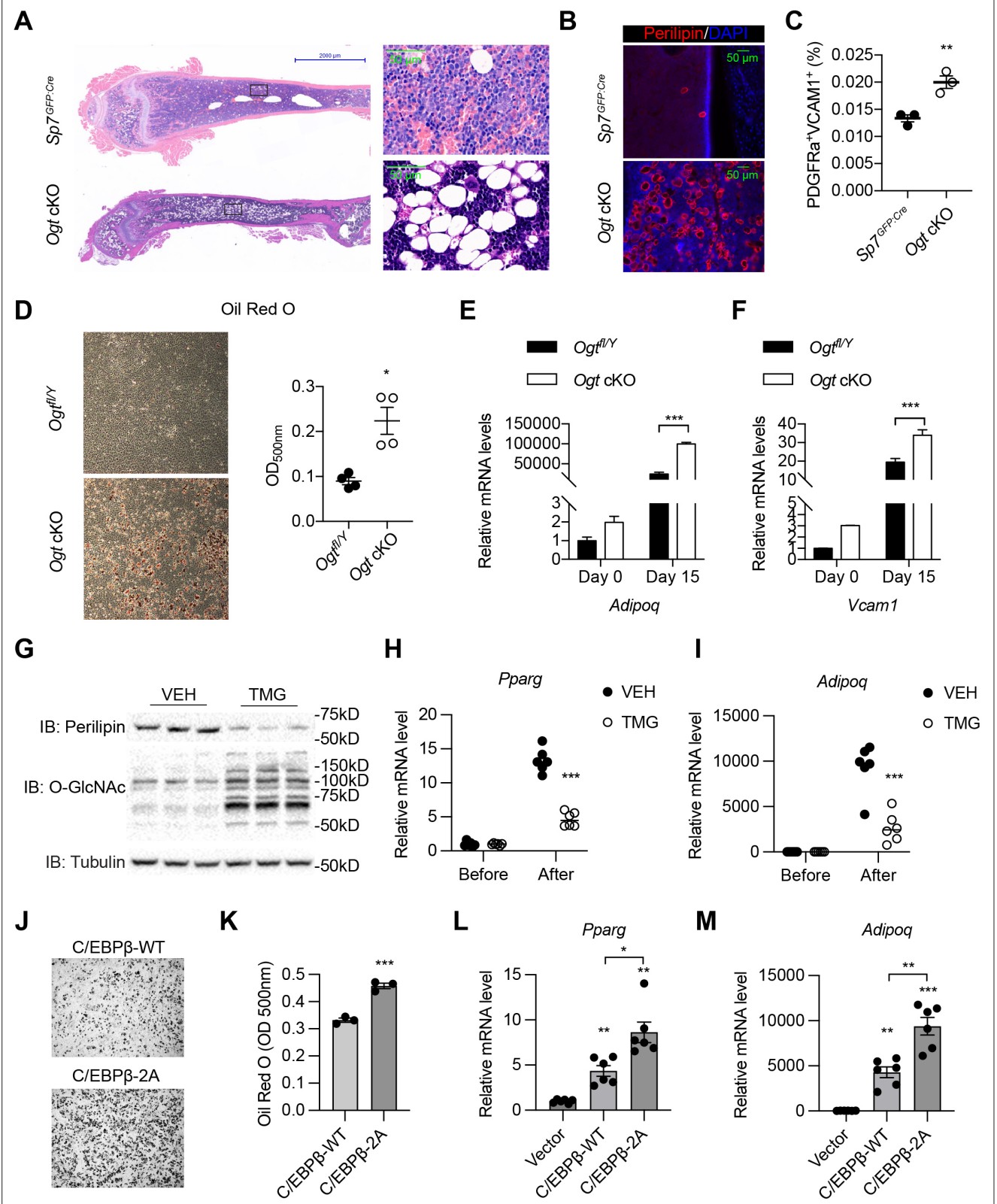

**Figure 4.** O-GlcNAcylation inhibits BM adipogenesis. (**A, B**) H&E (**A**) and Perilipin immunofluorescent staining (**B**) on femur sections from 4-week-old mice. (**C**) Flow cytometric quantification of PDGFRa⁺VCAM1⁺ preadipocytes frequencies within the live BM cells (n=3). (**D**) Adipogenic differentiation of primary BMSCs from control and *Ogt* cKO mice. Lipid was stained with Oil Red O and quantified to the right (n=4). (**E, F**) Primary BMSCs were osteogenic differentiated for 0 or 15 days. Expression of *Adipoq* (**E**) and *Vcam1* (**F**) genes was determined by RT-qPCR (n=3). (**G–I**) C3H10T1/2 cells,

*Figure 4 continued on next page*

*Figure 4 continued*

treated with or without TMG, were adipogenic differentiated. Western blotting for perilipin and O-GlcNAc of differentiated cells (**G**) and RT-qPCR for adipogenic marker *Pparg* (**H**) and *Adipoq* (**I**) expression. (**J–M**) Adipogenic differentiation of C3H10T1/2 cells infected with lentiviral C/EBPβ. Oil Red O was stained (**J**) and quantified (**K**). *Pparg* (**L**) and *Adipoq* (**M**) gene expression was determined by RT-qPCR. Data are presented as mean ± SEM.*, p<0.05; **, p<0.01; and ***, p<0.001 by unpaired student's t-test (**C, D, K**), one-way ANOVA (**L, M**), and two-way ANOVA (**E, F, H, I**).

The online version of this article includes the following source data and figure supplement(s) for figure 4:

**Source data 1.** Mass spectrometry search results of all protein modifications (Table S1) and PPARγ2 O-GlcNAc sites (Table S2).

**Source data 2.** Raw uncropped images for panel G.

**Figure supplement 1.** *Pdgra⁺Vcam1⁺* cells as adipogenic progenitors.

**Figure supplement 1—source data 1.** Raw uncropped images for *Figure 4—figure supplement 1B*.

**Figure supplement 2.** RUNX2 inhibits adipogenesis independently of O-GlcNAcylation.

**Figure supplement 3.** PPARγ2 O-GlcNAcylation is required for adipogenesis.

osteogenesis but does not suppress the chromatin remodeling needed for the activation of adipogenic transcriptional factors.

PPARγ1 is O-GlcNAcylated at T54 in the A/B activation domain (*Ji et al., 2012*), corresponding to T84 in PPARγ2, the major isoform in adipocytes (*Figure 4—figure supplement 3A*). Mutating T84 in PPARγ2 did not ablate the ability of the OGA inhibitor TMG to suppress adipogenesis in C3H10T1/2 cells (data not shown), suggesting the existence of other unidentified O-GlcNAc sites on PPARγ2 or other target proteins than PPARγ2. Through mass spectrometry, we were able to map four additional O-GlcNAc sites on PPARγ2 (*Figure 4—figure supplement 3A* and *Figure 4—source data 1*). Intriguingly, mutating these four sites or together with T84 to alanine, render PPARγ2 incompetent to induce transcription and adipogenesis (*Figure 4—figure supplement 3B, C*). It suggests that PPARγ2 O-GlcNAcylation is essential for adipocyte maturation, but likely does not mediate the anti-adipogenic effect of OGT in perinatal BMSCs.

We then looked to C/EBPβ, an early transcription factor that specifies the adipogenic fate of BMSCs (*Cao et al., 1991*; *Darlington et al., 1998*). It has been reported that OGT modifies C/EBPβ to inhibit its transcriptional activity (*Li et al., 2009*; *Qian et al., 2018*). As expected, ablating O-GlcNAcylation of C/EBPβ (2A mutation) promotes adipogenic differentiation of C3H10T1/2 cells (*Figure 4J and K*). Taken together, we concluded that, by O-GlcNAcylating and reciprocally regulating RUNX2 and C/EBPβ, OGT is required for the proper allocation of skeletal progenitors into osteogenic versus adipogenic lineages during development.

## OGT deficiency disrupts the BM niche

Skeletal development is concomitant with the establishment of definitive hematopoiesis in the BM. To test if OGT deficiency affects the niche function of *Sp7⁺* cells for B-cell lymphopoiesis, we performed flow cytometry analyses of bone marrow of 4-week-old mice (*Figure 5A*, *Figure 5—figure supplement 1* and *Figure 5—source data 1*; *Hardy et al., 1991*). No changes in the percentage of lineage⁻Sca-1⁺Kit⁺ (LSK) progenitor cells, common lymphoid progenitors (CLPs), Fraction A that contains pre-pro-B cells were observed between control and *Ogt* cKO mice (*Figure 5B–D*). While frequencies of Fraction B and C pro-B, pre-B, and immature B in *Ogt* cKO mice were drastically reduced (*Figure 5E–I*), demonstrating a developmental blockage from pre-pro-B to pro-B cells. In the peripheral blood, there was specific loss of CD19⁺B220⁺ B cells but not CD4⁺ or CD8⁺ T cells (*Figure 5J–L*). B-cell dysfunction observed here was similar to the phenotype in mice when all *Sp7⁺* cells were depleted (*Yu et al., 2016*) or IL-7 was deleted in BMSCs (*Cordeiro Gomes et al., 2016*), indicating that O-GlcNAcylation is essential for the *Sp7⁺* lineage cells to establish a niche environment for B-cell lymphopoiesis.

BM adiposity is associated with myeloid overproduction in conditions including aging, irradiation (*Ho et al., 2019*), osteopenia (*Kajkenova et al., 1997*), and obesity (*Singer et al., 2014*), indicating the supportive function of marrow adipocytes on demand-adapted myelopoiesis. Consistently with the increased BM adiposity found in *Ogt* cKO mice, we also observed biased HSPC differentiation toward the myeloid lineage, as shown by increased ratio of common myeloid progenitor (CMP) to common lymphoid progenitors (CLPs) and ratio of granulocyte-monocyte progenitors (GMP) to megakaryocyte-erythrocyte progenitors (MEP) in the BM (*Figure 5M and N*). As a result, increased numbers of red blood cells and neutrophils were observed in the blood of *Ogt* cKO mice (*Figure 5O*

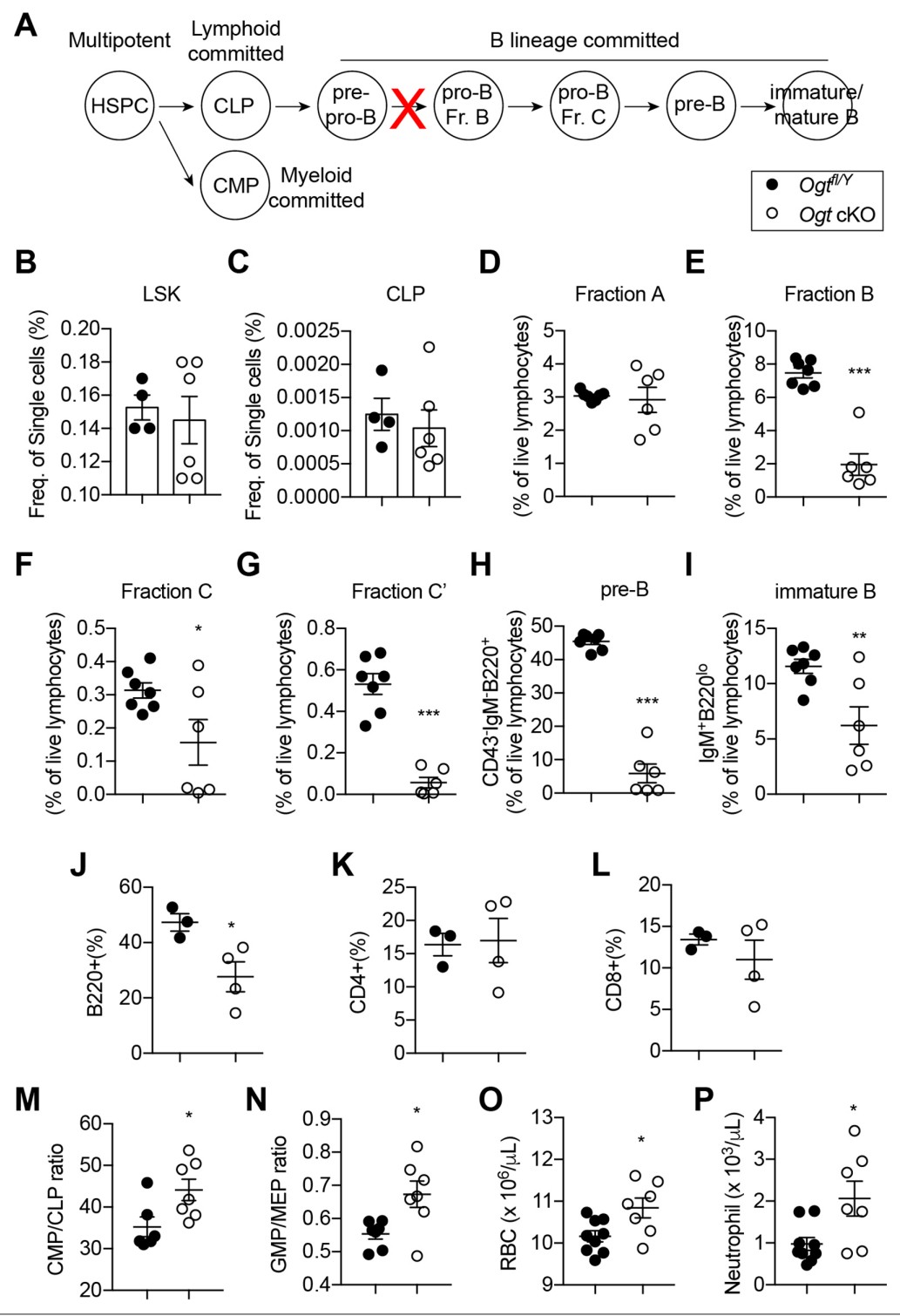

**Figure 5.** Impaired B lymphopoiesis and myeloid skewing in *Ogt* cKO mice. (**A**) Schematic view of B cell development in the BM and blockade by stromal OGT deficiency (red X). (**B–C**) Flow cytometric quantification of LSK (**B**) and CLP (**C**) among live BM cells (n=4–6). (**D–I**) Flow cytometric quantification of fraction A (**D**), fraction B (**E**), fraction C (**F**), fraction C' (**G**), fraction D (**H**), and immature B (**I**) frequencies among live BM lymphocytes (n=6–7). (**J–L**) Flow cytometric quantification of B220⁺ B cell (**J**), CD4⁺ T cell (**K**), and CD8⁺ T cell (**L**) percentages in the blood (n=3–4). (**M, N**) CMP/CLP ratio (**M**) and GMP/EMP ratio (**N**) in the BM (n=6–7). (**O, P**) Complete blood

*Figure 5 continued on next page*

*Figure 5 continued*

counting showing numbers of RBC (**O**) and neutrophil (**P**) (n=7–9). Data are presented as mean ± SEM. *, p<0.05; **, p<0.01; and ***, p<0.001 by unpaired student's t-test.

The online version of this article includes the following source data and figure supplement(s) for figure 5:

**Source data 1.** Antibodies used for flow cytometry.

**Figure supplement 1.** Flow cytometry of BM B cells.

*and P*). Together, these results demonstrate that OGT deficiency in neonatal BMSCs establishes a BM environment that promotes myelopoiesis and simultaneously impairs B cell development.

## Transcriptional regulation of niche cytokines by RUNX2 and C/EBPβ O-GlcNAcylation

BMSC-derived SCF (encoded by the *Kitl* gene) and IL-7 are required for the myeloid differentiation and B-cell development, respectively (*Asada et al., 2017*; *Cordeiro Gomes et al., 2016*; *Ding et al., 2012*). We sought to test if their expression is controlled by the same transcriptional network determining BMSC fate. Adipogenic differentiation of mesenchymal C3H10T1/2 cells concomitantly increased *Kitl* while decreased *Il7* gene expression (*Figure 6A and B*). Simultaneous treatment with the OGT inhibitor OSMI-1 dampened *Il7* expression before differentiation but enhanced *Kitl* expression in differentiated adipocytes (*Figure 6A and B*). On the other hand, osteogenic differentiation suppressed *Kitl* transcription, which could be further inhibited by TMG that elevated global O-GlcNAcylation (*Figure 6C*). While *Il7* mRNA levels were not evidently affected by osteogenic differentiation, TMG stimulated its expression (*Figure 6D*). O-GlcNAcylation inhibits the adipogenesis specified by C/EBPβ but supports osteogenesis determined by RUNX2. In concert, C/EBPβ overexpression in C3H10T1/2 cells activated *Kitl* transcription and suppressed *Il7* expression, which was further

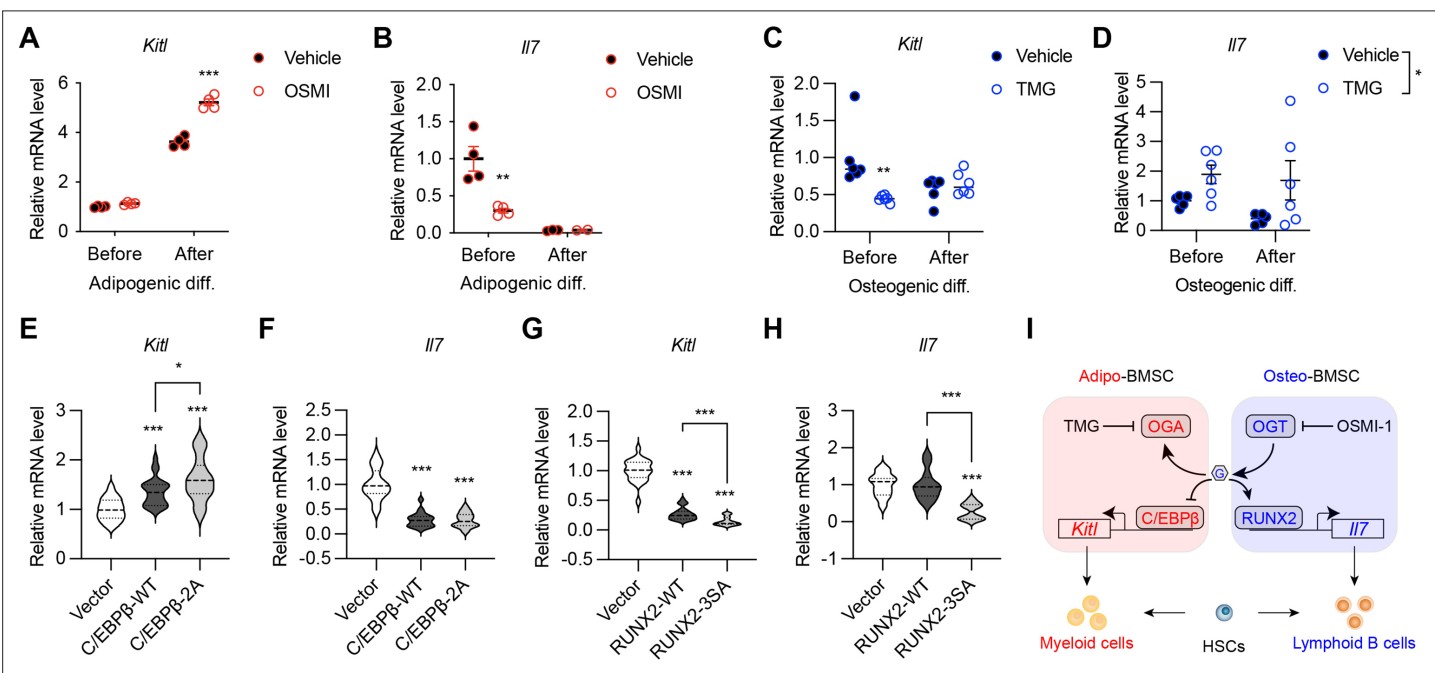

**Figure 6.** O-GlcNAc regulation of niche cytokine expression. (**A, B**) C3H10T1/2 cells were treated with vehicle or OGT inhibitor OSMI and differentiated for adipocytes (n=4). *Kitl* (**A**) and *Il7* (**B**) gene expression was determined by RT-qPCR. (**C, D**) C3H10T1/2 cells were treated with vehicle or OGA inhibitor TMG and induced for osteogenic differentiation (n=6). *Kitl* (**C**) and *Il7* (**D**) gene expression was determined by RT-qPCR. (**E–H**) C3H10T1/2 cells were infected with lentiviruses expressing WT and O-GlcNAc-deficient C/EBPβ (**E, F**) or RUNX2 (**G, H**). Expression *Kitl* (**E, G**) and *il7* (**F, H**) was measured by RT-qPCR (n=6). (**I**) Proposed action of protein O-GlcNAcylation in regulating the BMSC niche function. Data are presented as mean ± SEM. *, p<0.05; **, p<0.01; ***, p<0.001 by two-way ANOVA (**A–D**) and one-way ANOVA (**E–H**).

The online version of this article includes the following source data for figure 6:

**Source data 1.** Sequences of oligos used for RT-qPCR.

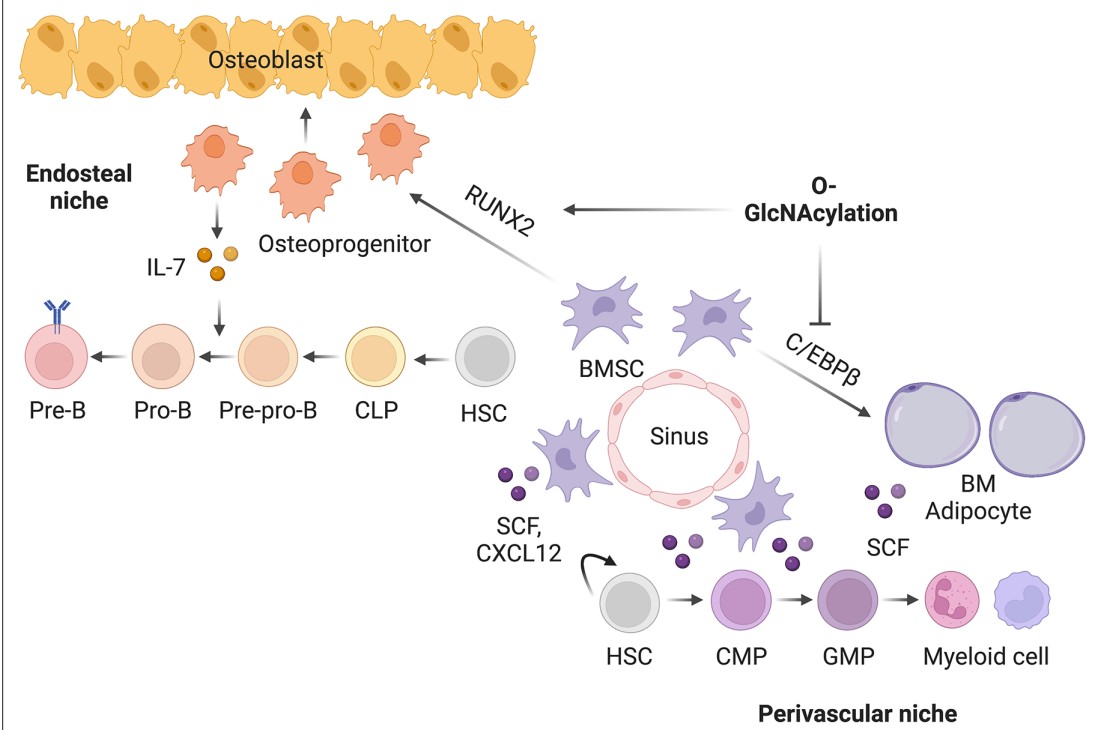

**Figure 7.** Working model of O-GlcNAc signaling in bone-BM development.

exacerbated by O-GlcNAc-deficient C/EBPβ (*Figure 6E and F*). However, RUNX2 overexpression decreased *Kitl* mRNA levels (*Figure 6G*). When compared to the wildtype, O-GlcNAc-defective RUNX2 was impaired in inducing *Il7* expression (*Figure 6H*). Collectively, these results reveal that protein O-GlcNAcylation, by acting on BMSC lineage transcriptional factors, establishes a pro-lymphopoietic niche during neonatal bone development and at the same time prevents the myeloid-skewing, adipogenic BM environment.

## Discussion

Post-translational modification networks exist in the bone-BM organ to regulate its development and remodeling. Given that definitive hematopoiesis is matured in perinatal BM, it is tempting to hypothesize that the regulatory mechanisms guiding the development of bone also establish the BM niche for hematopoiesis. However, experimental evidence has been largely lacking so far. In the present study, we examined the vital role of the under-studied protein O-GlcNAcylation in determining the osteogenic versus adipogenic fate specification of BMSCs and in balancing the pro-lymphopoietic and pro-myelopoietic niche function of BMSCs. We showed that, by modifying and reciprocally regulating RUNX2 and C/EBPβ, O-GlcNAc orchestrates the early development of skeletal and hematopoietic systems (*Figure 7*).

Multiple temporally and spatially distinct types of progenitors contribute to bone development and maintenance. In the early embryo, *Sp7*[+] progenitors give rise to fetal bone tissues and transient stromal cells that disappear in early postnatal life (*Mizoguchi et al., 2014*). Perinatally, *Sp7*[+] progenitors contribute to osteolineage cells and long-lived perivascular BMSCs that can be labeled by leptin receptor (*Lepr*) and adiponectin (*Adipoq*) (*Zhong et al., 2020*; *Zhou et al., 2017*). Recent evidence suggests that a significant portion of adult BMSCs and osteoblasts originate from collagen II (*Col2*)- and aggrecan (*Acan*)-expressing chondrocytes (*Ono et al., 2014*). Due to the fact that *Sp7*[GFP:Cre] targets osteoblasts, BMSCs, and a subset of chondrocytes (*Chen et al., 2014*; *Liu et al., 2013*), the current study could not delineate the exact developmental stages and the primary cellular compartments where OGT instructs bone development. Nonetheless, our ex vivo experiments and adult-onset targeting of OGT in *Sp7*[+] osteoblasts, together with prior published in vivo and in vitro

evidence (*Andrés-Bergós et al., 2012*; *Nagel and Ball, 2014*; *Nagel et al., 2013*), certainly reveal the indispensability of protein O-GlcNAcylation for chondro-osteogenic differentiation. While this study primarily focused early life bone development, it is warranted to further investigate the role of OGT in the transition to appositional remodeling during adulthood (*Shu et al., 2021*) and in osteoporosis pathogenesis during aging. Moreover, bone-forming skeletal stem cells (SSCs) are identified in other anatomical regions of long bones, such as growth plate, periosteum, and endosteum (*Ambrosi et al., 2019*). It remains undetermined whether O-GlcNAcylation is abundant in and controls the development and function of these SSC populations.

O-GlcNAcylation is required for PPARγ to drive adipogenesis, but why did not OGT-deficient BMSCs arrest their differentiation after being committed to the adipogenic lineage. One speculation is that PPARγ O-GlcNAcylation is extremely low in homeostatic conditions, which might help explain the rare appearance of adipocytes in young BM. If so, the loss of PPARγ O-GlcNAcylation in *Ogt* cKO mice would not block adipogenesis. Second, the differentiation of marrow adipose tissue (MAT) is distinct from peripheral white adipose tissue (WAT). For instance, adipogenic BMSCs in adult mice already express large amount of *Adipoq*, which is only present in mature WAT adipocytes. Certain forms of genetic lipodystrophy (e.g. mutations in *CAV1* and *PTRF*) selectively lose peripheral WAT but preserve MAT (*Scheller et al., 2015*). These findings suggest that MAT might be less dependent on PPARγ or able to adopt alternative differentiation when PPARγ is absent or inhibited. In fact, a recent publication reported a secondary adipogenic pathway in lipodystrophic 'fat-free' mice (*Zhang et al., 2021*). Lastly, our preliminary examination of old *Ogt* cKO mice revealed the resolution of BM adiposity, indicating that PPARγ and its O-GlcNAc modification become essential for the adipogenic differentiation of adult BMSCs.

Protein O-GlcNAcylation senses glucose availability (*Hardivillé and Hart, 2014*; *Ruan et al., 2012*), hormonal cues (*Ruan et al., 2014*; *Ruan et al., 2017*; *Whelan et al., 2008*), cellular stress (*Martinez et al., 2017*; *Ruan et al., 2017*), and immune signals (*Chang et al., 2020*; *Liu et al., 2019*; *Zhao et al., 2020*; *Zhao et al., 2022*) to maintain cellular and tissue homeostasis. Osteogenic differentiation of mesenchymal cells induces global O-GlcNAc levels (*Kim et al., 2007*; *Nagel and Ball, 2014*); however, the upstream mechanistic regulators of osteoblastic O-GlcNAcylation remain enigmatic. High glucose has been shown to promote O-GlcNAcylation and osteogenic differentiation of cartilage endplate stem cells (*Sun et al., 2019*). BMSCs preferentially use glycolysis for bioenergetics to support their self-renewal and multipotency (*Ito and Suda, 2014*; *van Gastel and Carmeliet, 2021*). Active aerobic glycolysis also fuels the high anabolic demand during bone formation. It would be important in the future to determine whether flux of the hexosamine biosynthetic pathway, a branch of glycolysis (*Ruan et al., 2013b*), increases to provide more UDP-GlcNAc for O-GlcNAc modification. We also showed here that PTH treatment increased protein O-GlcNAcylation. Signaling through the PTH receptor activates the cAMP-protein kinase A (PKA)-CREB pathway and the accumulation of inositol trisphosphate (IP3) and diacylglycerol (DAG), which further increase intracellular $Ca^{2+}$ and PKC, respectively (*Datta and Abou-Samra, 2009*). Future experiments are required to determine if OGT enzymatic activity can be regulated by these signaling nodes, for example $Ca^{2+}$/calmodulin-dependent protein kinase II (CaMKII) (*Ruan et al., 2017*). Sex differences in skeletal development, maintenance, and aging have been well appreciated. Whether BMSC O-GlcNAc signaling is differentially regulated between male and female animals, particularly during puberty and aging, is an important question that remains unaddressed. Since only male mice were investigated in the current study, it is unclear if the reciprocal regulation of RUNX2 and PPARγ by O-GlcNAcylation in determining the bone-fat balance is equally vital in females.

The BM microenvironment, composed of BMSCs, osteoblasts, adipocytes, sympathetic nerves, and vascular endothelial cells, has been highlighted as an important extrinsic factor for the maintenance and differentiation of distinct hematopoietic lineage progenitors (*Bianco and Robey, 2015*; *Calvi and Link, 2015*; *Morrison and Scadden, 2014*; *Wei and Frenette, 2018*). While the concomitant development, remodeling, and aging of the skeletal and hematopoietic systems have been observed in various pathophysiological conditions, mechanisms underlying the coordinated regulation of the two systems are less understood. Our current study has provided the first evidence that RUNX2, permitted by O-GlcNAcylation, not only is indispensable for the osteoblast development, but also establishes the endosteal niche for B lymphocytes by driving IL-7 expression (*Figure 7*). When OGT is deficient, the perivascular BMSCs are prone to adipogenic differentiation, which also activates

C/EBPβ-dependent SCF expression and myelopoiesis. During aging, the parallel dysfunction of the skeletal and hematopoietic systems leads to osteoporosis, marrow fat accumulation, and myeloid hematopoietic skewing (*Geiger et al., 2013*). Whether BMSC aging is associated with O-GlcNAc decline and whether the balance between RUNX and C/EBPβ leads to bone-fat imbalances and niche dysfunction require future investigations.

# Methods

**Key resources table**

| Reagent type (species) or resource | Designation | Source or reference | Identifiers | Additional information |
|---|---|---|---|---|
| Genetic reagent (*Mus musculus*) | *Sp7^GFP:Cre* (*B6.Cg-Tg(Sp7-tTA,tetO-EGFP/cre)1Amc/J*) | Jackson Laboratory | RRID:IMSR_JAX:006361 | |
| Genetic reagent (*Mus musculus*) | *Ogt^fl/fl* (*B6.129-Ogttm1Gwh/J*) | Jackson Laboratory | RRID:IMSR_JAX:004860 | |
| Cell line (*Mus musculus*) | Primary BMSC | This paper | | From long bones of mouse |
| Cell line (*Mus musculus*) | C3H10T1/2 | ATCC | CCL-226 | Verified by ATCC and tested negative for mycoplasma |
| Cell line (*Homo sapiens*) | HEK293FT | Invitrogen | R70007 | Verified by Invitrogen and tested negative for mycoplasma |
| Cell line (*Cercopithecus aethiops*) | COS7 | ATCC | CRL-1651 | Verified by ATCC and tested negative for mycoplasma |
| Transfected construct (*Mus musculus*) | 6xOSE2-luc | *Phimphilai et al., 2006* | | |
| Transfected construct (*Renilla reniformis*) | pGL4-hRluc | Promega | #E688A | |
| Transfected construct (*Mus musculus*) | PPREx3-TK-luc | Addgene | #1015 | |
| Antibody | Anti-Perilipin (Rabbit monoclonal) | Cell Signaling Technology | 9349T | IF(1:200) |
| Antibody | Anti-B220 (Rat monoclonal) | Life Technologies | 67-0452-82 | FC(1:200) |
| Antibody | Anti-CD43 (Rat monoclonal) | BD Biosciences | 553271 | FC(1:200) |
| Antibody | Anti-CD24 (Rat monoclonal) | Biolegend | 101822 | FC(1:1000) |
| Antibody | Anti-Ly-51 (Rat monoclonal) | Biolegend | 108305 | FC(1:200) |
| Antibody | Anti-CD127 (Rat monoclonal) | Tonbo Biosciences | 20–1271 U100 | FC(1:200) |
| Antibody | Anti-CD25 (Rat monoclonal) | Life Technologies | 63-0251-82 | FC(1:200) |
| Antibody | Anti-CD19 (Rat monoclonal) | Biolegend | 115545 | FC(1:200) |
| Antibody | Biotin-conjugated lineage antibodies (Rat monoclonal) | Biolegend | 133307 | FC(1:200) |
| Antibody | Anti-CD4 (Rat monoclonal) | Biolegend | 100403 | FC(1:200) |
| Antibody | Anti-CD5 (Rat monoclonal) | Biolegend | 100603 | FC(1:200) |
| Antibody | Anti-CD8 (Rat monoclonal) | Biolegend | 100703 | FC(1:200) |
| Antibody | Anti-CD127-APC (Rat monoclonal) | eBioscience | 17-1271-82 | FC(1:100) |
| Antibody | Anti-c-Kit-APC-eFluor780 (Rat monoclonal) | eBioscience | 47-1171-82 | FC(1:400) |

*Continued on next page*

*Continued*

| Reagent type (species) or resource | Designation | Source or reference | Identifiers | Additional information |
|---|---|---|---|---|
| Antibody | Anti-Sca-1-Super Bright 436 (Rat monoclonal) | eBioscience | 62-5981-82 | FC(1:100) |
| Antibody | Anti-CD34-PE (Rat monoclonal) | Biolegend | 152204 | FC(1:100) |
| Antibody | Anti-FcγR-PerCP-eFluor710 (Rat monoclonal) | eBioscience | 46-0161-80 | FC(1:400) |
| Antibody | Anti-CD150-BV605 (Rat monoclonal) | Biolegend | 115927 | FC(1:100) |
| Antibody | Anti-CD48-BUV395 (Rat monoclonal) | BD Biosciences | 740236 | FC(1:100) |
| Antibody | Anti-CD45-BUV395 (Rat monoclonal) | BD Biosciences | 564279 | FC(1:400) |
| Antibody | Anti-Ter119-BV421 (Rat monoclonal) | Biolegend | 116234 | FC(1:400) |
| Antibody | Anti-CD31-BV421 (Rat monoclonal) | Biolegend | 102424 | FC(1:400) |
| Antibody | Anti-PDGFRa-Super Bright 600 (Rat monoclonal) | eBioscience | 63-1401-82 | FC(1:100) |
| Antibody | Anti-VCAM1-PE (Rat monoclonal) | Biolegend | 105713 | FC(1:100) |
| Recombinant DNA reagent | RUNX2-WT | This paper | pLV-EF1a-RUNX2-WT-IRES-Hygro | See Methods; available upon request |
| Recombinant DNA reagent | RUNX2-3A/ RUNX2-3SA | This paper | pLV-EF1a-RUNX2-3Mut-IRES-Hygro | See Methods; available upon request |
| Recombinant DNA reagent | PPAR$\lambda$2-WT | This paper | pLVX- PPAR$\lambda$2-WT-Puro | See Methods; available upon request |
| Recombinant DNA reagent | PPAR$\lambda$2-T84A | This paper | pLVX- PPAR$\lambda$2-T84A-Puro | See Methods; available upon request |
| Recombinant DNA reagent | PPAR$\lambda$2-4A | This paper | pLVX- PPAR$\lambda$2-4A-Puro | See Methods; available upon request |
| Recombinant DNA reagent | PPAR$\lambda$2-5A | This paper | pLVX- PPAR$\lambda$2-5A-Puro | See Methods; available upon request |
| Recombinant DNA reagent | C/EBPβ-WT | This paper | pCDH-CMV-Cebpb-WT-P2a-Puro | See Methods; available upon request |
| Recombinant DNA reagent | C/EBPβ–2A | This paper | pCDH-CMV-Cebpb-2MUT-P2a-Puro | See Methods; available upon request |
| Commercial assay or kit | Q5 Site-Directed Mutagenesis Kit | NEB | #E0554 | |
| Commercial assay or kit | Dual-Luciferase Assay System | Promega | E1910 | |
| Commercial assay or kit | Transporter 5 Transfection Reagent | Polysciences | 26008–1 A | |
| Chemical compound, drug | Parathyroid hormone (PTH) | Genscript | RP01001 | |
| Chemical compound, drug | OSMI-1 | Sigma | SML1621-5MG | |
| Chemical compound, drug | Thiamet-G (TMG) | Biosynth | MD08856 | |
| Chemical compound, drug | Doxycycline food | Bio-Serv | S3888 | |
| Chemical compound, drug | IBMX | CAYMAN | 13347 | |
| Chemical compound, drug | Dexamethasone | Sigma | D4902 | |

*Continued on next page*

*Continued*

| Reagent type (species) or resource | Designation | Source or reference | Identifiers | Additional information |
|---|---|---|---|---|
| Chemical compound, drug | Insulin | Sigma | 91077 C | |
| Chemical compound, drug | Rosiglitazone | Sigma | R2408-10MG | |
| Chemical compound, drug | Ascorbic acid | Sigma | A4403-100MG | |
| Chemical compound, drug | β-Glycerophosphate | Santa cruz | sc-220452 | |
| Software, algorithm | FlowJ | BD Life Sciences | V10 | |

## Animals

All animal experiments were approved by the institutional animal care and use committee of the University of Minnesota (protocol # 2112–39682 A). All the mice were group-housed in light/dark cycle- (6am-8pm light), temperature- (21.5 ± 1.5 °C), and humidity-controlled (30–70%) room, and had free access to water and regular chow (Teklad #2018) unless otherwise indicated. Moist food was provided to constitutive $Sp7^{GFP:Cre}$ animals to circumvent tooth defects and prevent malnutrition. All mice were maintained on a C57BL6 background. Due to the X-chromosome localization of the $Ogt$ gene, only male mice were used in the study if not specified in the text or figures. To suppress Cre activity, designated breeders were fed a diet containing 200 mg/kg doxycycline (Bio-serv, S3888).

## BMSC isolation, culture, and differentiation

BMSC were isolated from the long bones as described previously (*Zhu et al., 2010*). The fragments of long bones were digested with collagenase II for 30 min. The released cells were discarded, and the digested bone fragments were cultivated in the BMSCs growth medium (alpha-MEM supplemented with 10% FBS). Once confluent, cells were switched to either adipogenic differentiation medium (alpha-MEM supplemented with 20% FBS, 500 µM IBMX, 1 µM Dexamethasone, 10 µg/ml Insulin and 1 µM Rosiglitazone) for the first 2 days. The medium was then changed to adipocyte differentiation base medium (α-MEM supplemented with 20% FBS, 10 µg/ml Insulin and 1 µM Rosiglitazone) for the next 4 days followed by oil red O staining. For osteogenic differentiation, cells were induced with osteoblast differentiation medium (α-MEM supplemented with 10% FBS, 0.3 mM ascorbic acid, 10 mM β-glycerophosphate, 0.1 µM Dexamethasone) for 14 days followed by ALP staining or for 28 days followed by Alizarin red staining.

## Cell culture, plasmids, and lentiviruses

HEK 293, COS7, and C3H10T1/2 (ATCC, CCL-226) cells were cultured with DMEM plus 10% of FBS. The mouse RUNX2-Myc/DDK plasmid was purchased from OriGene (MR227321), then subcloned into pLV-EF1a-IRES-Hygro (Addgene #85134). Mouse PPARγ2 with a N-terminal MYC tag was subcloned into pLVX-Dsred-puro plasmid. C/EBPβ plasmids were kindly provided by Dr. Xiaoyong Yang at Yale University and then subcloned into pCDH-CMV-P2a-Puro. O-GlcNAc sites were mutated into alanine with Q5 Site-Directed Mutagenesis Kit (NEB#E0554). Lentivirus was packed as previously described (*Huang et al., 2022*). Briefly, 293 FT cells were transfected with over-expression plasmids pSPAX2, and pMD2.G. Media with lentivirus were filtered and added into C3H10T1/2 cells. Seventy-two hr after infection, cells were then selected with drugs according to the resistance genes they possessed.

## Luciferase assay

For Runx2 luciferase assay, empty or RUNX2 vectors were transfected into COS7 cells with Lipofectamine, together with 6xOSE2-luc (*Phimphilai et al., 2006*) and pGL4-hRluc vectors in which either firefly or Renilla luciferase genes were expressed under the control of the RUNX2-specific or the constitutive SV40 promoter, respectively. After 6 hr, cells were washed three times and with the addition of 50 µM OSMI-1. Cells were incubated for an additional 48 hr in growth medium containing 5% serum. Luminescent signals were generated using the Dual-Luciferase Assay System (Promega). Relative light units (RLU) for the 6xOSE2 reporter were normalized against pGL4-hRLuc values as an internal control for transfection efficiency. For PPARγ2 luciferase assays, C3H10T1/2 cells were

transfected with Transporter 5 Transfection Reagent (Polysciences) following manufacture's protocol. PPARγ2 transcriptional activity was determined using the PPREx3-TK-luc reporter (Addgene, #1015).

## Histology

Bone tissues were fixed in formalin solution at 4 °C for 24 hr. Tissue embedding, sectioning, and hematoxylin and eosin staining were performed at the Comparative Pathology Shared Resource of the University of Minnesota. For immunostaining, the tissues were embedded in OCT then cut into 7 μm slides. After three times of PBS wash, the slides were incubated with blocking buffer (3% BSA in PBS) for 1 hr, then immersed with anti-Perilipin (Cell Signaling Technology, #9349) antibody overnight at 4 °C. For immunofluorescence, PBS-washed slides were incubated with a fluorescent secondary antibody at room temperature for 1 hr, and then mounted with VECTASHIELD Antifade Mounting Medium with DAPI after three times of PBS wash. A Nikon system was used for imaging. Goldner's trichrome and Safranin O staining were performed at Servicebio, China.

## micro-CT

The samples were scanned with an in vitro micro-CT device (Skyscan 1272, Bruker micro-CT) with scanning parameters of: Source Voltage = 60 kV, Source Current = 166 μA, exposure 897ms/frame, average of 3 frames per projection, Rotation Step (deg)=0.200 and 0.25 mm Aluminum filter. The specimens were scanned at high resolution (2016×1344 pixels) with an Isotropic voxel size of 7.1 μm. Reconstructions for X-ray projections and re-alignment were performed using the Skyscan software (NRecon and DataViewer) (v. 1.7.3.1, Brüker micro-CT, Kontich, Belgium). Ring artefact and beam hardening corrections were applied in reconstruction. Datasets were loaded into SkyScan CT-Analyzer software for measurement of BMD. Calibration was performed with 0.25- and 0.75 mg/mL hydroxy-apatite mice phantoms provided by SkyScan. For cancellous and cortical bone analysis, the scanning regions were confined to the distal metaphysis, 100 slices starting at 0.5 mm proximally from the proximal tip of the primary spongiosa for the cancellous portion and 100 slices starting at 4.5 mm proximally from the center of intercondylar fossa for the cortical portion.

## O-GlcNAc mass spectrometry

Myc-tagged PPARγ2 was co-transfected with OGT into 15 cm-dishes of 293T cells and purified by immunoprecipitation with anti-c-Myc agarose beads (Pierce), followed by PAGE gel electrophoresis. The corresponding PPARγ2 band was cut for in gel Trypsin (Promega) digestion. Tryptic peptides were analyzed by on-line LC-MS/MS using an Orbitrap Fusion Lumos (Thermo) coupled with a NanoAcquity UPLC system (Waters) as we previously reported (*Liu et al., 2019*; *Zhao et al., 2022*). Peaklists were generated using PAVA (UCSF) and searched using Protein Prospector 5.23.0 against the SwissProt database and a randomized concatenated database with the addition of the recombinant PPARγ2 sequence. HexNAcylated peptides were manually verified.

## Real-time RT-PCR

RNA was isolated with Trizol and reverse transcribed into cDNA with the iScript cDNA Synthesis Kit. Real-time RT-PCR was performed using iTaq Universal SYBR Green Supermix and gene-specific primers (*Figure 6—source data 1*) on a Bio-Rad C1000 Thermal Cycler.

## Flow cytometry

For PDGFRa+VCAM1+preadipocytes, BM cells were stained with anti-CD45, anti-Ter119, anti-CD31, anti-PDGFRa, and anti-VCAM1. We gated the PDGFRa+VCAM1+ cells after excluding the CD45+Ter119+CD31+ cells. For B-cell lymphopoiesis, BM cells were stained in PBS containing 1% (w/v) bovine serum albumin on ice for 30 min, with anti-B220 (Life Technologies, 67-0452-82), anti-CD43 (BD Biosciences, 553271), anti-CD24 (Biolegend, 101822), anti-Ly-51 (Biolegend, 108305), anti-CD127 (Tonbo Biosciences, 20–1271 U100), anti-CD25 (Life Technologies, 63-0251-82) and anti-CD19 (Biolegend, 115545). For hematopoietic stem and progenitor cells, BM cells were stained with a cocktail of biotin-conjugated lineage antibodies CD3e, B220, Ter119, Mac-1 and Gr-1 (Biolegend, 133307), CD4 (Biolegend, 100403), CD5 (Biolegend, 100603), CD8 (Biolegend, 100703), followed by Streptavidin-AF488 (Biolegend, 405235). Cells were then stained with CD127-APC (eBioscience, 17-1271-82), c-Kit-APC-eFluor780 (eBioscience, 47-1171-82), Sca-1-Super Bright 436 (eBioscience, 62-5981-82),

CD34-PE (Biolegend, 152204) and FcγR-PerCP-eFluor710 (eBioscience, 46-0161-80), CD150-BV605 (Biolegend, 115927), and CD48-BUV395 (BDBioscience, 740236). Fixable Viability Dye was used to exclude dead cells as instructed by the manufacturer. A complete list of used antibodies was shown in *Figure 5—source data 1*. Flow cytometry was performed on an LSR Fortessa H0081 or X20 and analyzed with FlowJo.

## Quantification and statistical analysis

Results are shown as mean ± SEM. N values (biological replicates) and statistical analysis methods are described in figure legends. The statistical comparisons were carried out using two-tailed unpaired Student's t-test and one-way or two-way ANOVA with indicated post hoc tests with Prism 9 (Graphpad). Differences were considered significant when $p < 0.05$; *, $p < 0.05$; **, $p < 0.01$; ***, $p < 0.001$.

## Acknowledgements

We thank Dr. Xiaoyong Yang for kindly providing wildtype and mutant C/EBPβ plasmids; Dr. Renny Franchesci for providing the 6xOSE2-luc plasmid; and Alexis Nagel for Runx2 luciferase assay. This work was supported by the National Natural Science Foundation of China, China (32170847) to ZH, NIH/NIAID (R01 AI162678) to HZ, the Miriam and Sheldon G Adelson Medical Research Foundation to JCM and ALB, and NIH/NIAID (R01 AI139420 and R01 AI162791) to H-BR.

## Additional information

### Funding

| Funder | Grant reference number | Author |
|---|---|---|
| National Natural Science Foundation of China | 32170847 | Zan Huang |
| National Institutes of Health | R01 AI162678 | Hu Zeng |
| National Institutes of Health | R01 AI139420 | Hai-Bin Ruan |
| National Institutes of Health | R01 AI162791 | Hai-Bin Ruan |
| Dr. Miriam and Sheldon G. Adelson Medical Research Foundation | | Jason C Maynard<br>Alma L Burlingame |

The funders had no role in study design, data collection and interpretation, or the decision to submit the work for publication.

### Author contributions

Zengdi Zhang, Formal analysis, Investigation, Methodology, Writing - original draft; Zan Huang, Formal analysis, Funding acquisition, Investigation, Methodology, Writing - original draft; Mohamed Awad, Jason C Maynard, Data curation, Investigation, Methodology; Mohammed Elsalanty, Supervision, Methodology; James Cray, Resources, Methodology; Lauren E Ball, Investigation; Alma L Burlingame, Kim C Mansky, Resources, Supervision; Hu Zeng, Supervision, Funding acquisition, Investigation; Hai-Bin Ruan, Conceptualization, Resources, Supervision, Funding acquisition, Writing - original draft, Writing - review and editing

### Author ORCIDs

Hai-Bin Ruan (iD) http://orcid.org/0000-0002-3858-1272

### Ethics

All animal experiments were approved by the institutional animal care and use committee of the University of Minnesota (protocol # 2112-39682A).

Decision letter and Author response

Decision letter https://doi.org/10.7554/eLife.85464.sa1

Author response https://doi.org/10.7554/eLife.85464.sa2

## Additional files

### Supplementary files

• MDAR checklist

### Data availability

All data generated or analyzed during this study are included in the manuscript and supporting file. Supplementary tables have been provided for Mass spectrometry, primer sequences, and antibody list.

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
