## [Editor Report]

Bone marrow stromal cells (BMSCs) can differentiate into a variety of cell types such as osteoblasts, chondrocytes, and adipocytes. The authors of this important study provide compelling and strong evidence that ablating O-GlcNAc transferase (OGT) in BMSCs impairs bone formation but promotes marrow adiposity. The results show that the balance of osteogenic and adipogenic differentiation of BMSCs is controlled by reciprocal O-GlcNAc regulation of lineage-specifying transcription factors, and highlights the importance of an intracellular glycosylation process of specific proteins in establishing the BM niche for hematopoiesis.

---

## [Decision Letter]

**Decision letter after peer review:**

Thank you for submitting your article "O-GlcNAc glycosylation orchestrates fate decision and niche function of bone marrow stromal progenitors" for consideration by *eLife*. Your article has been reviewed by 3 peer reviewers, and the evaluation has been overseen by a Reviewing Editor and Marianne Bronner as the Senior Editor. The following individuals involved in the review of your submission have agreed to reveal their identity: Tamas L Horvath (Reviewer #1); Kosaku Shinoda (Reviewer #2).

Essential revisions:

There is no need to provide new data. In order to improve the manuscript, please substantially address the reviewers' questions or concerns to revise the manuscript. Meanwhile, please provide point-to-point responses to the editors' and reviewers' comments.

*Reviewer #1 (Recommendations for the authors):*

In my judgment, this paper is appropriate for publication. I would request that the authors detail the results in relation to males versus females in each experiment.

*Reviewer #2 (Recommendations for the authors):*

I will mainly comment on the biology of adipocytes, which is my specialty.

The point that O-GlcNAc regulates adipocyte differentiation has been convincingly demonstrated, but the molecular mechanism has not been fully explored. The data that C/EBPβ increases transcriptional activity by mutating O-GlcNAc sites, but does this happen independently of protein level, or is it due to altered protein stability (ubiquitination)? Also, downstream of O-GlcNAcylated C/EBPβ, does Pparg expression change at the protein level, and is there a change in its interaction with the binding partner, C/EBPalpha? If these points are clarified, this would be an excellent paper that could be laid out more generally in adipocyte research.

I also have a concern about the PDGFRa+VCAM1+ population of adipocyte progenitors, which the authors have shown to be unregulated by approximately 30% by knockout of Ogt (Figure 4C). How developmentally significant this difference is? Looking at panel A of the same figure, the phenotype seems to be a more qualitative difference, but can a 30% difference in PDGFRa+ explain it? Rather, could this be through turning on the expression of Pparg, which indicates lineage commitment, be more important? Sca-1 is commonly used as a marker for adipogenic MSCs – authors should cite a paper showing that PDGFRa+VCAM1+ is a selective marker for adipogenic cells and a detailed protocol for the FACS should also be added in the Materials and methods section. For example, which lineage markers were used to deplete non-adipogenic cells?

Finally, what regulates O-GlcNAcylation of C/EBPβ? Does aging or a Western diet regulate this PTM? Pursuing this will strengthen the significance of this novel regulatory layer of adipogenesis.

---

## [Author Response]

Reviewer #1 (Recommendations for the authors):In my judgment, this paper is appropriate for publication. I would request that the authors detail the results in relation to males versus females in each experiment.

I appreciate Reviewer #1’s enthusiasm and recommendation on manuscript. Thankyou.

Reviewer #2 (Recommendations for the authors):I will mainly comment on the biology of adipocytes, which is my specialty.The point that O-GlcNAc regulates adipocyte differentiation has been convincingly demonstrated, but the molecular mechanism has not been fully explored. The data that C/EBPβ increases transcriptional activity by mutating O-GlcNAc sites, but does this happen independently of protein level, or is it due to altered protein stability (ubiquitination)? Also, downstream of O-GlcNAcylated C/EBPβ, does Pparg expression change at the protein level, and is there a change in its interaction with the binding partner, C/EBPalpha? If these points are clarified, this would be an excellent paper that could be laid out more generally in adipocyte research.

These are great points. We have Western blotting data (Author response image 1) demonstrating that O-GlcNAc mutant C/EBPβ did not change its protein expression. This is consistent with our previous study in other cell types (Qian et al., 2018). We have not determined if protein stability or ubiquitination are affected, because we did not see any change in its protein level. Regarding PPARγ, we observed increased gene expression when mutant C/EBPβ was expressed in C3H10T_1/2_ cells (Figure 4H), suggesting adipogenic activation. We did not measure PPARγ protein levels or its interaction with C/EBPα, the latter of which is an important question and will be addressed in our future experiments.

**Author response image 1. sa2fig1:** 

I also have a concern about the PDGFRa+VCAM1+ population of adipocyte progenitors, which the authors have shown to be unregulated by approximately 30% by knockout of Ogt (Figure 4C). How developmentally significant this difference is? Looking at panel A of the same figure, the phenotype seems to be a more qualitative difference, but can a 30% difference in PDGFRa+ explain it? Rather, could this be through turning on the expression of Pparg, which indicates lineage commitment, be more important? Sca-1 is commonly used as a marker for adipogenic MSCs – authors should cite a paper showing that PDGFRa+VCAM1+ is a selective marker for adipogenic cells and a detailed protocol for the FACS should also be added in the Materials and methods section. For example, which lineage markers were used to deplete non-adipogenic cells?

We agree with the reviewer that the 30% increase in PDGFRa^+^VCAM1^+^ adipose progenitors cannot fully explain the BM adiposity observed in *Ogt* cKO animals. A simultaneous augmentation in adipogenic differentiation is most likely. Indeed, we provided in vitro evidence that genetic deletion or chemical inhibition of OGT significant increases adipogenesis (Figure 4D-I). With scRNA-sequencing, our unpublished data and publicly available datasets by others (Baryawno et al., 2019; Zhong et al., 2020) have repeatedly showed that *Pdgra*^+^/*Vcam1*^+^ doulble positive cells overlap with *Lepr*^+^/*Adipoq*^+^ adipogenic progenitors (also known as MALP and Adipo-CAR cells in some publications). On the other hand, the *Ly6a* gene (encoding Sca-1) marks a more primitive population of mesenchymal stem cells (Zhong *et al.*, 2020). Expression of these genes at the single cell level are now provided in Figure 4—figure supplement 1**.** To increase rigor, we have now provided detailed flow cytometry protocol (including negative lineage markers) in the Methods section.

Finally, what regulates O-GlcNAcylation of C/EBPβ? Does aging or a Western diet regulate this PTM? Pursuing this will strengthen the significance of this novel regulatory layer of adipogenesis.

Those are all great questions. The short answer is that we do not know yet (as we briefly discussed in the 3^rd^ paragraph in the Discussion section). The fact that PTH is sufficient to induce protein O-GlcNAcylation (Figure 2D) would support the prediction that aging perturbs C/EBPβ O-GlcNAcylation and the bone-fat balance. In fact, we are actively investigating these possibilities using the *Lepr^Cre^* (instead of *Sp7^GFP:Cre^*) mice to target OGT in adult perivascular BMSCs. We are excited about the preliminary data we have collected so far and expect to report our findings in very near future.

Reference:

Baryawno, N., Przybylski, D., Kowalczyk, M.S., Kfoury, Y., Severe, N., Gustafsson, K., Kokkaliaris, K.D., Mercier, F., Tabaka, M., Hofree, M., et al. (2019). A Cellular Taxonomy of the Bone Marrow Stroma in Homeostasis and Leukemia. Cell *177*, 1915-1932 e1916. 10.1016/j.cell.2019.04.040.

Chen, J., Shi, Y., Regan, J., Karuppaiah, K., Ornitz, D.M., and Long, F. (2014). Osx-Cre targets multiple cell types besides osteoblast lineage in postnatal mice. PLoS One *9*, e85161. 10.1371/journal.pone.0085161.

Ji, S., Park, S.Y., Roth, J., Kim, H.S., and Cho, J.W. (2012). O-GlcNAc modification of PPARgamma reduces its transcriptional activity. Biochem Biophys Res Commun *417*, 1158-1163. 10.1016/j.bbrc.2011.12.086.

Li, M.D., Vera, N.B., Yang, Y., Zhang, B., Ni, W., Ziso-Qejvanaj, E., Ding, S., Zhang, K., Yin, R., Wang, S., et al. (2018). Adipocyte OGT governs diet-induced hyperphagia and obesity. Nat Commun *9*, 5103. 10.1038/s41467-018-07461-x.

Qian, K., Wang, S., Fu, M., Zhou, J., Singh, J.P., Li, M.D., Yang, Y., Zhang, K., Wu, J., Nie, Y., et al. (2018). Transcriptional regulation of O-GlcNAc homeostasis is disrupted in pancreatic cancer. J Biol Chem *293*, 13989-14000. 10.1074/jbc.RA118.004709.

Rodda, S.J., and McMahon, A.P. (2006). Distinct roles for Hedgehog and canonical Wnt signaling in specification, differentiation and maintenance of osteoblast progenitors. Development *133*, 3231-3244. 10.1242/dev.02480.

Scheller, E.L., Doucette, C.R., Learman, B.S., Cawthorn, W.P., Khandaker, S., Schell, B., Wu, B., Ding, S.Y., Bredella, M.A., Fazeli, P.K., et al. (2015). Region-specific variation in the properties of skeletal adipocytes reveals regulated and constitutive marrow adipose tissues. Nat Commun *6*, 7808. 10.1038/ncomms8808.

Yang, Y., Fu, M., Li, M.D., Zhang, K., Zhang, B., Wang, S., Liu, Y., Ni, W., Ong, Q., Mi, J., and Yang, X. (2020). O-GlcNAc transferase inhibits visceral fat lipolysis and promotes diet-induced obesity. Nat Commun *11*, 181. 10.1038/s41467-019-13914-8.

Zhang, X., Robles, H., Magee, K.L., Lorenz, M.R., Wang, Z., Harris, C.A., Craft, C.S., and Scheller, E.L. (2021). A bone-specific adipogenesis pathway in fat-free mice defines key origins and adaptations of bone marrow adipocytes with age and disease. e*Life 10*. 10.7554/*eLife*.66275.

Zhong, L., Yao, L., Tower, R.J., Wei, Y., Miao, Z., Park, J., Shrestha, R., Wang, L., Yu, W., Holdreith, N., et al. (2020). Single cell transcriptomics identifies a unique adipose lineage cell population that regulates bone marrow environment. e*Life 9*. 10.7554/*eLife*.54695.